# Association between Endometriosis and Delivery Outcomes: A Systematic Review and Meta-Analysis

**DOI:** 10.3390/biomedicines10020478

**Published:** 2022-02-17

**Authors:** Yoshikazu Nagase, Shinya Matsuzaki, Yutaka Ueda, Mamoru Kakuda, Sahori Kakuda, Hitomi Sakaguchi, Michihide Maeda, Tsuyoshi Hisa, Shoji Kamiura

**Affiliations:** 1Department of Obstetrics and Gynecology, Osaka University Graduate School of Medicine, Osaka 565-0871, Japan; ynagase@gyne.med.osaka-u.ac.jp (Y.N.); mamorukakuda@gyne.med.osaka-u.ac.jp (M.K.); 2Department of Gynecology, Osaka International Cancer Institute, Osaka 541-8567, Japan; kakuda-sa@mc.pref.osaka.jp (S.K.); hitomi.sakaguchi@oici.jp (H.S.); michihide.maeda@oici.jp (M.M.); hisa-tu@mc.pref.osaka.jp (T.H.); kamiura-sh@oici.jp (S.K.)

**Keywords:** cesarean delivery, delivery outcomes, endometriosis, postpartum hemorrhage, systematic review

## Abstract

Endometriosis is a common benign gynecological disorder; however, delivery outcomes concerning pregnancies with endometriosis remain understudied. This study aimed to assess the effect of endometriosis on delivery outcomes, including the rate of instrumental delivery, cesarean delivery (CD), postpartum hemorrhage (PPH), and perioperative complications during CD. A systematic literature review was conducted using multiple computerized databases, and 28 studies met the inclusion criteria. Pooled analysis showed that histologically diagnosed endometriosis was associated with an increased rate of instrumental delivery (odds ratio [OR] 1.26, 95% confidence interval [CI] 1.02–1.56) and an increased risk of CD (adjusted OR 2.59, 95%CI 1.32–5.07). In our analysis, histologically diagnosed endometriosis was not associated with an increased rate of PPH; however, one retrospective study reported that endometriosis increased the rate of PPH during CD (adjusted OR 1.7, 95%CI 1.5–2.0). Two studies examined perioperative complications during CD, and women with deep endometriosis had a higher rate of bowel resection or bladder injury than those without endometriosis. Our findings suggest that endometriosis is a significant risk factor for instrumental delivery and CD and may be associated with a higher rate of PPH and intraoperative complications during CD.

## 1. General Overview

Endometriosis is a benign gynecological disorder characterized as the presence of endometrial glands and stroma in areas other than the luminal surface or myometrium of the uterus, causing chronic inflammation and symptoms such as dysmenorrhea, pelvic adhesions, and infertility [1,2,3]. Endometriosis causes infertility due to abnormal follicle formation, oxidative stress, impaired immune function, and decreased endometrial receptivity [4,5,6]. The number of patients with endometriosis has been reported to account for approximately 5–10% of reproductive-aged women and approximately 5–50% of infertile women [7,8,9,10].

Chronic pelvic inflammation due to endometriosis causes adhesions between the posterior surface of the uterus and ovaries, rectum, and pelvic peritoneum, resulting in cul-de-sac obliteration or posterior extrauterine adhesion (PEUA) [11,12]. We hypothesized that chronic inflammation and adhesions due to endometriosis may be correlated with adverse delivery outcomes, such as increased rates of cesarean delivery (CD), instrumental delivery, and postpartum hemorrhage (PPH). Regarding surgical complications during gynecological surgery, intrapelvic adhesions due to endometriosis contribute to prolonged surgery time and increased complications [13,14]. Based on these findings, endometriosis could increase surgical complications during CD.

Several systematic reviews and retrospective studies that have examined the effects of endometriosis on perinatal complications have reported a risk of increased preterm birth and placenta previa [15,16]. However, the relationship between endometriosis and delivery outcomes, including the rate of instrumental delivery, CD, and PPH, is understudied. Therefore, providing evidence-based information concerning the risk of complications during delivery in women with endometriosis is likely to help inform and guide clinical practice.

To investigate delivery outcomes in women with endometriosis, a systematic review was performed based on the results of previous studies using multiple computerized databases. The primary aim of this review was to assess the effect of endometriosis on delivery outcomes, including the risk of instrumental delivery, CD, PPH, and perioperative complications during CD. The secondary aim was to investigate the techniques, suggestions, and new initiatives for improved deliveries in pregnant women with endometriosis. If a study on the outcome of interest was not identified in this systematic review, a narrative review was added for further consideration.

## 2. Materials and Methods

### 2.1. Approach for the Systematic Literature Review

A systematic review was conducted to determine the effects of endometriosis on delivery outcomes (rate of instrumental delivery, CD, PPH, and surgical complications during CD), as well as to explore the techniques, suggestions, and new initiatives for improved deliveries in pregnant women with endometriosis. We undertook this systematic literature review in accordance with the Preferred Reporting Items for Systematic Reviews and Meta-Analyses, i.e., PRISMA guidelines [17].

A systematic search was performed using PubMed, Scopus, and the Cochrane Central Register of Controlled Trials from their inception to 15 November 2021, using MeSH terms, if applicable, and text words for the concepts “Endometriosis”, “Pregnancy” and “Delivery” (Appendix A) as previously undertaken [15,18,19,20]. There were no restrictions on date, language, or other parameters. Two review authors (Y.N. and S.M.) identified relevant studies through screening their titles and abstracts. This systematic review was not pre-registered.

### 2.2. Definition of Endometriosis in Previous Studies

Since accurate diagnosis of endometriosis during pregnancy is challenging, endometriosis during pregnancy was defined in this study as previously described [21]: (i) a past surgical history for endometriosis; (ii) a clinical diagnosis of endometriosis (ultrasonographic findings, pelvic adhesions, PEUA, and ovarian endometrioma); (iii) a clinical or histopathological diagnosis of endometriosis during CD; and (iv) an International Classification of Diseases code registration for endometriosis.

### 2.3. Study Selection and Data Extraction

Studies were included in this review if they (i) were comparative studies between an exposed group (pregnant women with endometriosis) and a control group (pregnant women without endometriosis); (ii) examined the effect of endometriosis on the rate of instrumental delivery, CD, and PPH; (iii) compared surgical complications during CD between gravid patients with and without endometriosis; and (iv) included techniques and suggestions for improved deliveries in women with endometriosis. Studies were excluded from this review if they (i) had insufficient information concerning the outcome of interest; (ii) lacked a control arm; (iii) did not meet this study’s definition of endometriosis; (iv) had not been written in English; and (v) were case reports, case series, reviews, systematic reviews, and meta-analyses.

The authors (Y.N. and S.M.) extracted the variables, study details (author names, publication year, and study location), number of included cases, definition and types of endometriosis, and outcomes of interest (rate of instrumental delivery, CD and PPH).

### 2.4. Analysis of Outcome Measures and Assessment of Risk of Bias

In this systematic review and meta-analysis, two sensitivity analyses were performed, focusing on the diagnostic criteria (histological or non-histological diagnosis) and types of endometriosis (deep endometriosis [DE] or non-DE). Risk of bias assessment was performed using the Risk of Bias in Non-randomized Studies-of Interventions tool (ROBINS-I) as previously described [22,23,24].

### 2.5. Meta-Analysis Plan

From eligible studies, the risks in terms of the outcomes of interest were computed and presented as odds ratios (ORs) with 95% confidence intervals (CIs). Heterogeneity across the studies was examined using *I*^2^ statistics to measure the percentage of total variation [25].

The meta-analysis and image production were performed using RevMan v5.4.1 software (Cochrane Collaboration, Copenhagen, Denmark). For consistency, the data from all outcomes (continuous and bivariate) were entered into this software in such a way that negative effect sizes or relative risks <1 favored active intervention. In the pooled analysis, if studies had low heterogeneity (*I*^2^: <30%), a fixed-effect analysis was applied. If studies had moderate heterogeneity (*I*^2^: 30–60%) to considerable heterogeneity (*I*^2^: 75–100%), random-effect analysis was conducted.

### 2.6. Statistical Analysis

Fisher’s exact or chi-square tests were used to analyze differences in baseline demographics between the two groups. Statistical significance was set at *p* < 0.05. Analyses were performed using Statistical Package for Social Sciences (IBM SPSS, v28.0, Armonk, NY, USA) software.

## 3. Results

### 3.1. Results of the Systematic Review

#### 3.1.1. Study Selection and Characteristics

The study selection scheme is shown in Figure 1. In the literature search, 2674 studies were examined and 28 studies, comprising 92,418 women with endometriosis and 4,626,840 women without endometriosis, met the inclusion criteria of this systematic literature review [26,27,28,29,30,31,32,33,34,35,36,37,38,39,40,41,42,43,44,45,46,47,48,49,50,51,52,53]. The metadata of the included studies are summarized in Appendix A. Twenty-eight of the eligible studies were retrospective studies, and none were randomized controlled trials. The studies included in this systematic review had been published between 2003 and 2021 in Europe (*n* = 16 [57.1%]) [26,30,32,34,38,39,40,42,43,44,45,46,48,50,52,53], China (*n* = 4 [14.3%]) [27,28,37,47], Japan (*n* = 3 [10.7%]) [31,41,49], Israel (*n* = 2 [7.1%]) [33,36], Australia (*n* = 1 [3.6%]) [51], Canada (*n* = 1 [3.6%]) [35], and Korea (*n* = 1 [3.6%]) [29].

Of the 28 eligible studies, nine studies examined the effect of endometriosis on the rate of instrumental delivery (Table 1) [31,32,33,34,38,39,44,48,53]. Twenty-seven studies investigated the effect of endometriosis on the risk of CD (Table 2) [26,27,28,29,30,31,32,33,34,35,36,37,38,39,40,41,42,43,44,45,46,47,48,49,50,52,53], of which six studies conducted sensitivity analyses focusing on types of endometriosis (Table 3) [32,34,40,44,46,50]. Furthermore, 17 studies examined the rate of PPH between pregnant women with and without endometriosis (Table 4) [27,29,30,31,32,33,34,35,36,37,38,39,40,42,45,48,51]. The risk of perioperative complications during CD was evaluated in two studies [44,46].

#### 3.1.2. Risk of Bias in the Included Studies

The risk of bias assessment for the comparative studies is shown in Appendix A. Of the studies assessed (*n* = 28), moderate publication bias (moderate quality) in 23 studies and severe publication bias (low quality) in the other 5 studies were observed.

#### 3.1.3. Definitions of Endometriosis and PPH

Of the 28 included studies, patients with endometriosis were histologically diagnosed in 15 studies [27,28,32,34,36,37,38,39,40,44,46,47,48,49,53]. Of the 17 studies that examined the association between endometriosis and PPH, nine studies clearly stated the definition of PPH. However, the diagnostic criteria for PPH varied among these studies (Table 4) [30,31,33,34,39,40,42,48,51].

### 3.2. Results of the Meta-Analysis

#### 3.2.1. Primary Outcome: Rate of Instrumental Delivery

Nine studies with moderate quality examined the effect of endometriosis on the rate of instrumental delivery [31,32,33,34,38,39,44,48,53], of which seven included patients with histologically diagnosed endometriosis (Table 1) [32,34,38,39,44,48,53]. We performed a random-effect analysis due to substantial heterogeneity. In the unadjusted pooled analysis, the rate of instrumental delivery was similar between the endometriosis and non-endometriosis groups (*n* = 9; OR 1.06, 95%CI 0.81–1.38; heterogeneity: *p* < 0.01, *I*^2^ = 85%; Figure 2A). In the unadjusted pooled analysis, histologically diagnosed endometriosis was correlated with an increased rate of instrumental delivery (*n* = 7; OR 1.26, 95%CI 1.02–1.56; heterogeneity: *p* < 0.01, *I*^2^ = 70%; Figure 2B).

#### 3.2.2. Primary Outcome: The CD Rate

Of 28 included studies, 27 (5 of low and 22 of moderate quality) examined the association between endometriosis and the risk of CD (Table 2) [26,27,28,29,30,31,32,33,34,35,36,37,38,39,40,41,42,43,44,45,46,47,48,49,50,52,53]. A random-effect analysis was conducted because of considerable heterogeneity. The unadjusted pooled analysis (*n* = 27) indicated that pregnant women with endometriosis were more likely to have a higher CD rate than those without endometriosis (OR 1.91, 95%CI 1.60–2.28; heterogeneity: *p* < 0.01, *I*^2^ = 99%; Figure 3A). The adjusted pooled analysis (*n* = 11) showed results similar to those of the unadjusted analysis (OR 1.57, 95%CI 1.39–1.78; heterogeneity: *p* < 0.01, *I*^2^ = 92%; Figure 3B).

Furthermore, we conducted pooled analyses restricted to 15 studies investigating the effect of histologically diagnosed endometriosis on CD rates [27,28,32,34,36,37,38,39,40,44,46,47,48,49,53]. In a sensitivity analysis that focused on histologically diagnosed endometriosis, women with endometriosis were found to be more likely to have an increased CD rate compared with those without endometriosis in both the unadjusted (*n* = 15; OR 2.30, 95%CI 1.52–3.49; heterogeneity: *p* < 0.01, *I*^2^ = 98%; Figure 4A) and adjusted pooled analyses (*n* = 4; OR 2.59, 95%CI 1.32–5.07; heterogeneity: *p* < 0.01, *I*^2^ = 90%; Figure 4B).

A sensitivity analysis was conducted to investigate the relationship between endometriosis types and the risk of CD. Of six studies that described endometriosis types, five included patients with DE [32,34,40,44,46] and one included patients with ovarian endometrioma [50] (Table 3). In the unadjusted pooled analyses, pregnant women with DE had an increased risk of CD compared with those without endometriosis (*n* = 5; OR 2.36, 95%CI 1.72–3.23; heterogeneity: *p* = 0.6, *I*^2^ = 0%; Figure 5A). Moreover, pregnant women with non-DE had an increased CD rate compared with those without endometriosis (Figure 5B) (*n* = 3; OR 1.51, 95%CI 1.07–2.13; heterogeneity: *p* = 0.17, *I*^2^ = 43%). In a comparison of CD rates between those with DE and those with non-DE, the CD rate was similar between the two groups (*n* = 3; OR 1.11, 95%CI 0.66–1.86; heterogeneity: *p* = 0.88, *I*^2^ = 0%; Figure 5C).

#### 3.2.3. Co-Primary Outcome: Rate of PPH

Of 28 included studies, 17 (4 of low and 13 of moderate quality) examined the rate of PPH between pregnant women with and without endometriosis (Table 4) [27,29,30,31,32,33,34,35,36,37,38,39,40,42,45,48,51]. A random-effect analysis was used because of considerable heterogeneity. The unadjusted pooled analysis (*n* = 17) showed that pregnant women with endometriosis were at a higher risk of PPH than those without endometriosis (OR 1.22, 95%CI 1.03–1.43; heterogeneity: *p* < 0.01, *I*^2^ = 93%; Figure 6A). The adjusted pooled analysis (*n* = 8) showed results similar to those of the unadjusted analysis (OR 1.14, 95%CI 1.01–1.28; heterogeneity: *p* < 0.01, *I*^2^ = 86%; Figure 6B).

In sensitivity analyses focusing on histology-confirmed endometriosis, nine studies were included to investigate the effect of histologically diagnosed endometriosis on the rate of PPH [27,32,34,36,37,38,39,40,48]. In this analysis, women with histologically diagnosed endometriosis were found to be more likely to have an increased rate of PPH in the unadjusted analysis (*n* = 9; OR 1.57, 95%CI 1.47–1.67; heterogeneity: *p* < 0.01, *I*^2^ = 77%; Figure 7A), whereas this association was not observed in the adjusted analysis (*n* = 3; OR 1.24, 95%CI 0.81–1.89; heterogeneity: *p* < 0.01, *I*^2^ = 93%; Figure 7B).

To explore the risk of PPH according to the mode of delivery (vaginal or CD), a sub-analysis of the included studies was performed. In this analysis, one population-based study that examined the risk of PPH in pregnant women with endometriosis in terms of mode of delivery was included. In that study, primiparous women with endometriosis had a higher risk of PPH after CD than those without endometriosis (2.3% [191/8190] vs. 1.4% [6362/447,574], adjusted OR 1.7, 95%CI 1.5–2.0), whereas this relationship was not observed in women after vaginal delivery (6.7% [550/8190] vs. 8.5% [37,978/447,574], adjusted OR 0.9, 95%CI 0.8–0.9) [38].

#### 3.2.4. Co-Primary Outcome: Surgical Complications during CD

Only two retrospective studies reported perioperative complications during CD in pregnant women with endometriosis [44,46]. Of these studies, the first study reported that women with DE were at a high risk of surgical complications (hysterectomy, 7.1% [2/28]; bowel resection, 3.6% [1/28]; and bladder injury, 7.1% [2/28]) [44]. The second study reported that one of 18 (5.6%) patients with DE had a bladder injury requiring reconstructive urological surgery during CD [46]. These results suggest that DE may be associated with an increased rate of intraoperative complications during CD.

#### 3.2.5. Secondary Outcome: Techniques, Suggestions, and New Initiatives for Improved Deliveries in Pregnant Women with Endometriosis

Our systematic literature review identified only one study that suggested management practices for patients with placenta previa and PEUA due to endometriosis [12]. This study was a retrospective comparative analysis of patients with placenta previa with and without endometriosis, which we reported in 2020. Using propensity score matching, we reported that intraoperative blood loss was significantly higher in an endometriosis group (*n* = 24) than in a non-endometriosis group (*n* = 48) (1515 mL vs. 870 mL, *p* < 0.01). That study showed that patients with placenta previa and endometriosis may be at a higher risk of PPH compared with those without endometriosis.

Moreover, this study proposed the following CD-related suggestions for patients with placenta previa and endometriosis: (i) minimization of uterine exteriorization and adhesion detachment, and (ii) in women with PPH, application of the Bakri balloon with Nelaton catheters to guide the cervical passage. The authors considered that this policy and practice would be useful in decreasing intraoperative blood loss in women with placenta previa and PEUA. In fact, intraoperative blood loss has been reported to have significantly decreased in these patients following the introduction of these types of measures (1180 mL vs. 1827 mL, *p* = 0.02), when compared with intraoperative blood loss in patients who had been treated prior to implementing such measures [12].

## 4. Discussion

### 4.1. Principal Findings

The principal findings of this study were as follows. First, our systematic review and meta-analysis showed that endometriosis increased the instrumental delivery and CD rates. Second, the rate of PPH after CD may increase in pregnant women with endometriosis. Third, while further studies on the relationship between endometriosis and perioperative complications are needed, women with DE are at risk of surgical complications such as bladder injury during CD.

### 4.2. Comparison with Existing Literature

#### 4.2.1. Primary Outcome: Rate of Instrumental Delivery and CD

The results of this systematic review showed that no studies similar to this study had been published concerning the relationship between endometriosis and instrumental delivery, and that histologically diagnosed endometriosis significantly increased the risk of instrumental delivery.

Several meta-analyses have reported the effects of endometriosis on the CD rate, and our study results were similar to the findings of those studies [9,54,55]. However, no meta-analysis has evaluated the risk of CD through focusing on diagnostic criteria and endometriosis types. Our sensitivity analyses showed that endometriosis increased the risk of CD, regardless of its diagnostic criteria or types. Two studies included in this systematic review contained detailed information concerning the indications for CD in women with endometriosis. An increased CD rate due to an increased frequency of breech presentation, placenta previa, and dystocia was suggested [33,34].

Our study findings indicated that pregnant women with endometriosis had an increased rate of instrumental delivery and CD, suggesting an association between endometriosis and dystocia. In two studies focusing on functional changes in the uterus due to endometriosis, it was suggested that endometriosis and endometriotic pelvic adhesions lead to uterine dysperistalsis and reduced contractility [56,57]. In addition, adhesions such as PEUA cause anatomical distortion and reduce contractility of the uterus, resulting in an incompatible orientation between the uterus and fetus, which may lead to failure to progress labor [33,34].

#### 4.2.2. Co-Primary Outcome: Rate of PPH

Histologically diagnosed endometriosis increased the risk of PPH in the unadjusted analysis but not in the adjusted analysis. However, the studies involved in the pooled adjusted analysis were very limited, and this analysis is inadequate for determining the relationship between endometriosis and PPH. A systematic review and meta-analysis on the association between endometriosis and the risk of PPH has previously been conducted [15], and the results of this study were similar to those of our previous study. However, in our previous study, we focused on a more restricted range of literature in relation to the risk of endometriosis on the rate of placenta previa. To address that study’s limitations, this study examined the effects of endometriosis on the rate of PPH in a broader range of studies, involving the inclusion all studies that had investigated the effect of endometriosis on delivery outcomes, which allowed for a more accurate assessment of the effects of endometriosis on the risk of PPH.

The findings of this analysis indicated that endometriosis increased the rate of instrumental delivery and CD; therefore, the risk of PPH in patients with endometriosis according to the mode of delivery (vaginal or cesarean) should be examined. In a population-based study conducted in Denmark, primiparous women with endometriosis were reported to be more likely to have PPH after CD than those without endometriosis; however, this association was not observed in women after vaginal delivery [38]. Based on these findings, we speculate that (i) endometriosis may not affect uterine contractions because the rate of PPH was not increased during vaginal delivery, and (ii) CD in women with endometriosis may be challenging due to endometriosis and endometriotic adhesions. Only one of the 28 included studies had evaluated the risk of PPH in patients with ovarian endometrioma whose endometriotic lesions were confined to the ovaries [32]. That study reported that the risk of PPH was similar between patients with and without ovarian endometrioma (15.6% [10/64] vs. 24.4% [413/1690], *p* = 0.13) [32]. These findings from only one report need to be supplemented through further research to fully determine the association between ovarian endometrioma and PPH.

#### 4.2.3. Surgical Complications during CD

Although several studies have examined the effects of endometriosis on the CD rate, only two studies have reported perioperative complications during CD. These studies were retrospective Italian studies of patients with DE. A study by Exacoustos et al. reported that of 28 patients with DE who had undergone CD, one had a bowel resection, whereas two had bladder injuries [44]. In a second study by Baggio et al., one of 18 patients with DE had a bladder injury during CD [46]. Bladder and ureteral injuries are the most common intraoperative complications during CD, and pelvic adhesions due to a history of CD or endometriosis are major risk factors [58,59]. Therefore, we consider that further suggestions are needed concerning CD in patients with endometriosis to prevent intraoperative complications.

#### 4.2.4. A Proposed Surgical Technique and Suggestions for CD in Patients with Endometriosis

To our knowledge, there has been one single institutional retrospective study that has proposed a surgical technique and suggestions for CD in patients with placenta previa and PEUA [10]. That study comprised 24 women with PEUA, and the causes of PEUA were listed as follows: 12 women with suspected endometriosis during CD, 7 women who had undergone prior surgery for endometriosis, and 5 who were unable to be assessed as the uterine exteriorization was difficult.

A recommendation of that retrospective study was to insert a Bakri intrauterine balloon, using a modified Matsubara Nelaton method, without avoiding exteriorization of the uterus for women with PPH. The Matsubara Nelaton method was originally reported by Matsubara et al. as a safe technique for the insertion of an intrauterine tamponade balloon during CD in patients with placenta previa who had massive bleeding during CD [60]. Because the uterus is strongly retro-flexed in pregnant women with PEUA, it is often challenging to place an intrauterine balloon during CD. Therefore, the authors proposed intrauterine balloon tamponade using Nelaton catheters to guide cervical passage as a hemostatic procedure.

The use of a modified Matsubara Nelaton method has previously been reported to be associated with reducing intraoperative bleeding in pregnant women with placenta previa and PEUA [12]. In that study, in which only 24 women with PEUA were included, we considered the number of women to be insufficient to draw robust conclusions concerning the usefulness of the modified Matsubara Nelaton method. Therefore, further consideration is needed to manage the delivery of patients with endometriosis more effectively.

### 4.3. Strengths and Limitations

To our knowledge, this is the first systematic review and meta-analysis to investigate the relationship between endometriosis and delivery outcomes. Furthermore, sensitivity analyses focusing on diagnostic criteria and types of endometriosis have not been conducted in previous studies. These sensitivity analyses enhanced the robustness of the results in this study.

However, this study had some notable limitations. First, all studies included in this review were retrospective studies with differing diagnostic criteria for endometriosis and PPH and different indications for instrument delivery and CD. Therefore, the severe heterogeneity in the included studies may have led to a severe bias that readers should be aware of when interpreting the results of this meta-analysis.

Second, the systematic review protocol was not registered. Without preregistration, it is not known whether the main outcomes, such as the CD rate, had been predefined as primary outcomes. Therefore, bias could have been introduced into this systematic review.

Third, a recent comprehensive review reported that advanced maternal age was one risk factor for PPH [61]. Other risk factors for PPH include obstetric complications such as placenta previa and maternal complications such as uterine myoma. Many of the studies in this systematic review used an adjusted OR for maternal age, but not for various other confounding factors of PPH. Therefore, this study cannot clearly conclude that endometriosis is a risk factor for PPH as other potential confounding factors had been excluded. Similar to PPH, we were also unable to exclude confounding factors in relation to CD from the analysis; therefore, results of this study regarding the effect of endometriosis on the CD rate should be interpreted with caution.

Fourth, only two studies that focused on perioperative complications during CD in women with endometriosis were identified, which provided insufficient data to assess surgical outcomes during CD. Further studies are warranted to examine the effects of endometriosis on the surgical outcomes of CD in women with endometriosis. Since randomized controlled studies may be challenging, a multicenter study with a large sample size and unified diagnostic criteria for endometriosis and PPH is needed.

Fifth, while the effect of DE on surgical outcomes during CD was investigated in two studies, no studies have determined the effect of non-DE on such outcomes. Further studies are warranted to investigate the effects of endometriosis on CD-related surgical outcomes according to endometriosis types.

## 5. Conclusions and Implications

### 5.1. Implications for Practice

This systematic review and meta-analysis showed that endometriosis increased the rate of instrumental delivery and CD. Although the effects of endometriosis on the risk of PPH require further elucidation, there is a possibility that endometriosis is associated with an increased rate of PPH during CD. Based on the results of our systematic review, clinicians should consider the possibility of endometriosis in pregnant women. Since a diagnosis of endometriosis during pregnancy is challenging [62], a medical consultation would be helpful to determine whether endometriosis is present or whether there has been any past surgical history of endometriosis prior to pregnancy.

### 5.2. Implications for Clinical Research

Women with endometriosis are at high risk of CD and PPH; however, few studies have focused on related surgical techniques, suggestions, and new initiatives for improving CD in such patients. Further consideration is necessary to determine a more effective management course for deliveries in women with endometriosis.

## Figures and Tables

**Figure 1 biomedicines-10-00478-f001:**
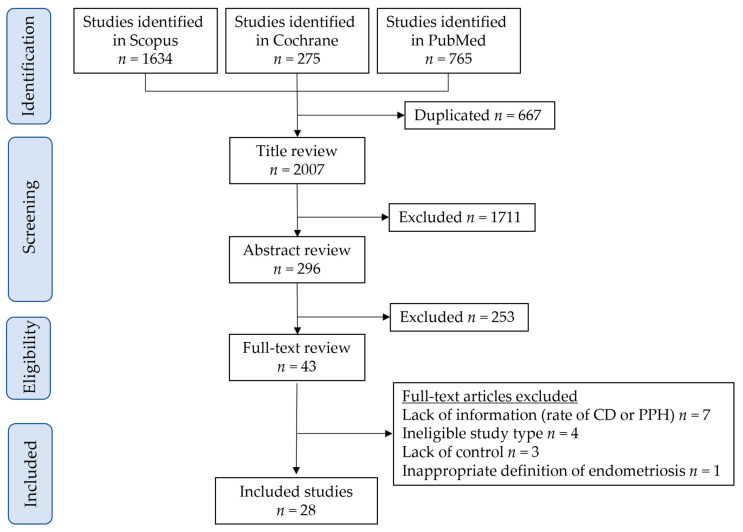
Study selection scheme for the systematic literature review. Abbreviations: CD, cesarean delivery; PPH, postpartum hemorrhage.

**Figure 2 biomedicines-10-00478-f002:**
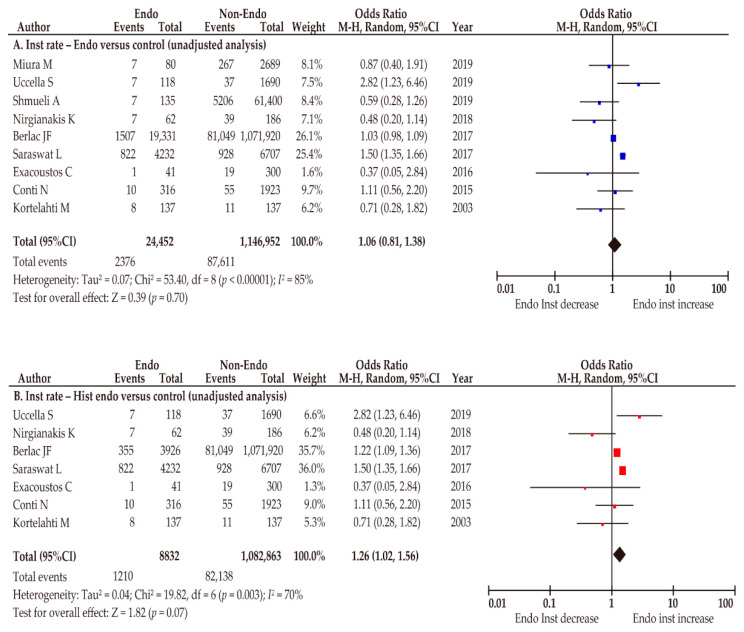
Meta-analysis of the effect of endometriosis on the rate of instrumental delivery. The pooled unadjusted OR for instrumental delivery (**A**) women with endometriosis vs. women without endometriosis and (**B**) women with histologically diagnosed endometriosis vs. women without endometriosis are shown. The forest plots were modified within the stratum according to year of publication and relative weight (%) of the study. Heterogeneity was substantial in both analyses ((**A**), *I*^2^ = 85%; (**B**), *I*^2^ = 70%). Some values listed might differ slightly from the original values because the calculation was performed using Revman v5.4.1. Abbreviations: CI, confidence interval; Endo, endometriosis; Hist endo, histologically diagnosed endometriosis; Inst, instrumental delivery, OR, odds ratio.

**Figure 3 biomedicines-10-00478-f003:**
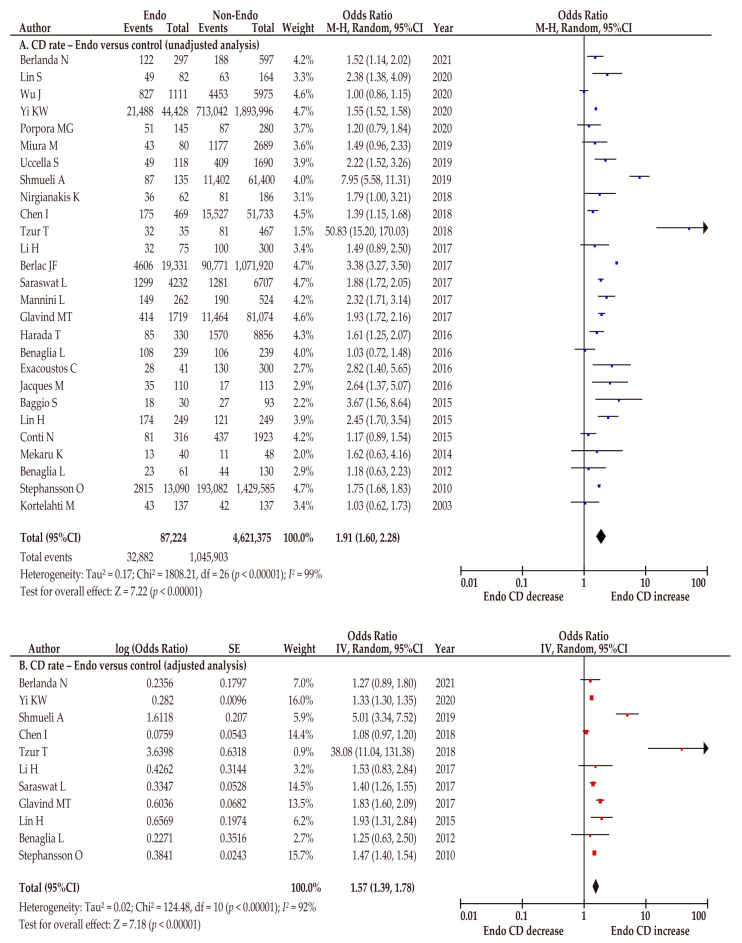
Meta-analysis of the effect of endometriosis on the CD rate. The pooled unadjusted (**A**) and adjusted (**B**) odds ratios for CD between women with and without endometriosis are shown. The forest plots were modified within the stratum according to year of publication and relative weight (%) of the study. Heterogeneity was considerable in both analyses ((**A**), *I*^2^ = 99%; (**B**), *I*^2^ = 92%). Some values listed might differ slightly from the original values because the calculation was performed using Revman v5.4.1. Abbreviations: CD, cesarean delivery; CI, confidence interval; Endo, endometriosis.

**Figure 4 biomedicines-10-00478-f004:**
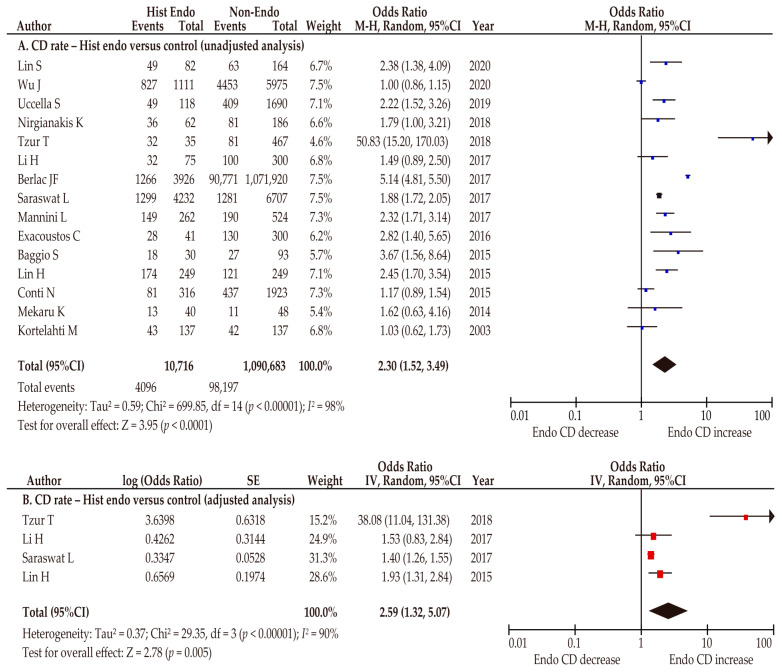
Meta-analysis of the effect of histologically diagnosed endometriosis on the CD rate. The pooled unadjusted (**A**) and adjusted (**B**) odds ratios for CD between women with histologically diagnosed endometriosis and women without endometriosis are shown. The forest plots were modified within the stratum according to year of publication and relative weight (%) of the study. Heterogeneity was considerable in both analyses ((**A**), *I*^2^ = 98%; (**B**), *I*^2^ =90%). Some values listed might differ slightly from the original values because the calculation was performed using Revman v5.4.1. Abbreviations: CD, cesarean delivery; CI, confidence interval; Endo, endometriosis; Hist endo, histologically diagnosed endometriosis.

**Figure 5 biomedicines-10-00478-f005:**
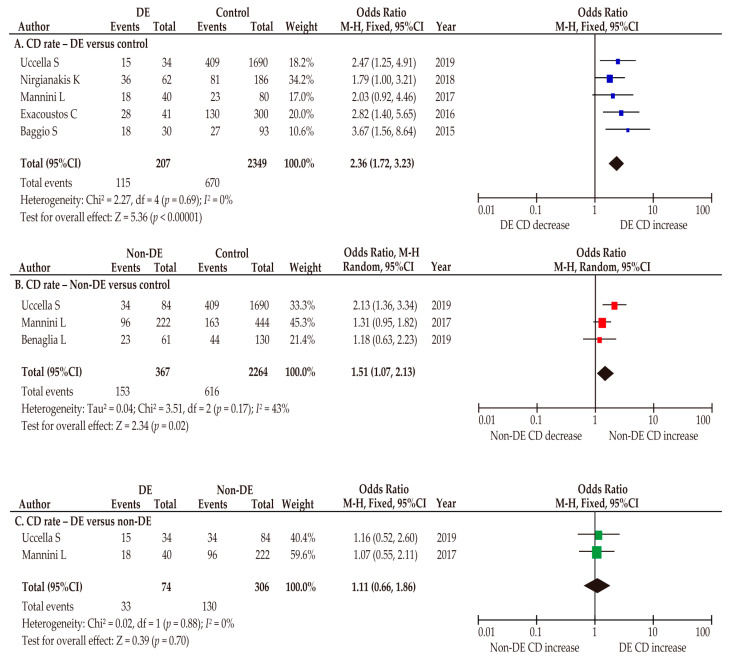
Meta-analysis of the effect of DE and non-DE on the CD rate. The pooled unadjusted odds ratio for CD (**A**) women with DE vs. women without endometriosis, (**B**) women with non-DE vs. women without endometriosis, and (**C**) women with DE vs. women with non-DE are shown. The forest plots were modified within the stratum according to year of publication and relative weight (%) of the study. Heterogeneity was low to moderate in three analyses ((**A**), *I*^2^ = 0%; (**B**), *I*^2^ = 43%; (**C**), *I*^2^ = 0%). Some values listed might differ slightly from the original values because the calculation was performed using Revman v5.4.1. Abbreviations: CD, cesarean delivery; CI, confidence interval; DE, deep endometriosis.

**Figure 6 biomedicines-10-00478-f006:**
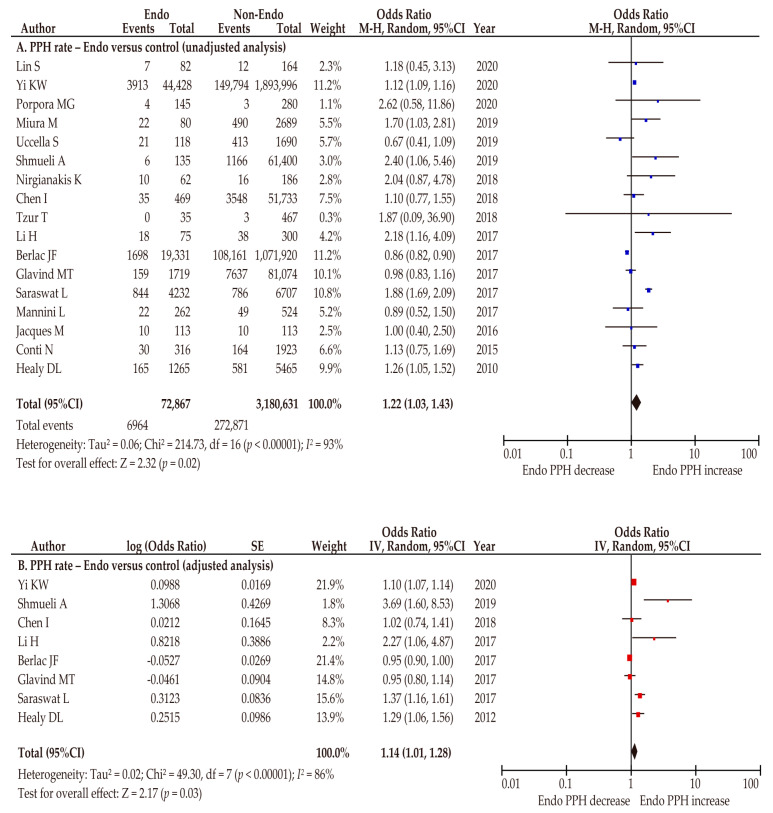
Meta-analysis of the effect of endometriosis on the rate of postpartum hemorrhage. The pooled unadjusted (**A**) and adjusted (**B**) odds ratios for postpartum hemorrhage between women with and without endometriosis are shown. The forest plots were modified within the stratum according to year of publication and relative weight (%) of the study. Heterogeneity was considerable in both analyses ((**A**), *I*^2^ = 93%; (**B**), *I*^2^ = 86%). Some values listed might differ slightly from the original values because the calculation was performed using Revman v5.4.1. Abbreviations: CI, confidence interval; Endo, endometriosis; PPH, postpartum hemorrhage.

**Figure 7 biomedicines-10-00478-f007:**
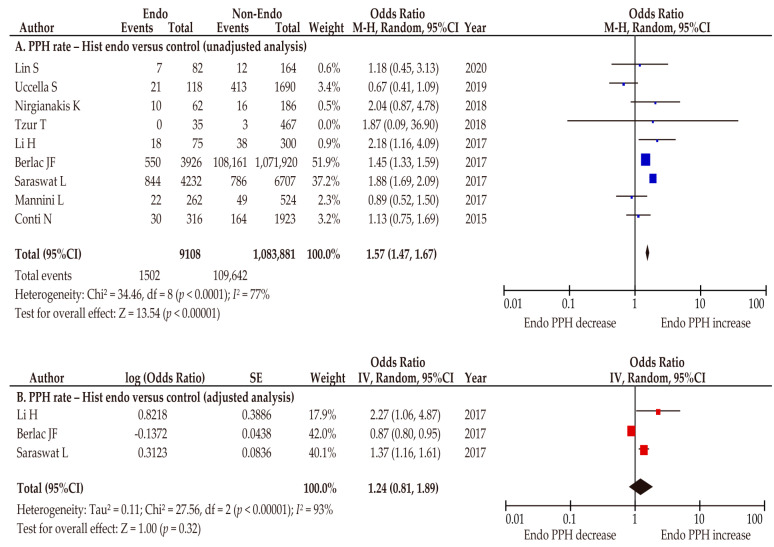
Meta-analysis of the effect of histologically diagnosed endometriosis on the rate of postpartum hemorrhage. The pooled unadjusted (**A**) and adjusted (**B**) odds ratios for postpartum hemorrhage between women with histologically diagnosed endometriosis and women without endometriosis are shown. The forest plots were modified within the stratum according to year of publication and relative weight (%) of the study. Heterogeneity was considerable in both analyses ((**A**), *I*^2^ = 77%; (**B**), *I*^2^ = 93%). Some values listed might differ slightly from the original values because the calculation was performed using Revman v5.4.1. Abbreviations: CI, confidence interval; Endo, endometriosis; Hist endo, histologically diagnosed endometriosis; PPH, postpartum hemorrhage.

**Table 1 biomedicines-10-00478-t001:** Comparator analysis of the rates of instrumental delivery between women with and without endometriosis.

Author	Year	Total No.	Endo No.	Control No.	Instrumental Delivery		Definition of Endo
					OR (95%CI)	Adjusted OR (95%CI)	
Miura M [31]	2019	2769	80	2689	0.87 (0.40–1.91)	-	Clinical or surgical diagnosis
Uccella S [32]	2019	1808	118	1690	2.82 (1.23–6.46)	-	Past surgical history
Shmueli A [33]	2019	61,535	135	61,400	0.59 (0.28–1.26)	-	ICD-10 code
Nirgianakis K [34]	2018	248	62	186	0.48 (0.20–1.14)	-	Past surgical history
Berlac JF [38]	2017	1,091,251	19,331	1,071,920	1.03 (0.98–1.09)	1.2 (1.1–1.3)	ICD-10 code
		1,075,846	3926 *	1,071,920	1.22 (1.09–1.36)	1.6 (1.5–1.8)	Past surgical history
Saraswat L [39]	2017	10,939	4232	6707	1.50 (1.35–1.66)	1.21 (1.08–1.36)	Past surgical history
Exacoustos C [44]	2016	341	41	300	0.37 (0.05–2.84)	-	Past surgical history
Conti N [48]	2015	2239	316	1923	1.11 (0.56–2.20)	-	Past surgical history
Kortelahti M [53]	2003	274	137	137	0.71 (0.28–1.82)	-	Past surgical history

* Restricted to women with histologically diagnosed endometriosis. Some values listed might differ slightly from the original values because the calculation was performed using Revman v5.4.1. Some values listed might differ slightly from the original values, as estimated by the authors. Abbreviations: CI, confidence interval; Endo, endometriosis; ICD-10, International Classification of Diseases 10th Revision; No., number of cases; OR, odds ratio; -, not applicable.

**Table 2 biomedicines-10-00478-t002:** Comparator analysis of the rates of CD between women with and without endometriosis.

Author	Year	Total No.	Endo No.	Control No.	CD		Definition of Endo
					OR (95%CI)	Adjusted OR (95%CI)	
Berlanda N [26]	2021	894	297	597	1.52 (1.14–2.02)	1.27 (0.89–1.80)	Clinical or surgical diagnosis
Lin S [27]	2020	246	82	164	2.38 (1.38–4.09)	-	Past surgical history
Wu J [28]	2020	7086	1111	5975	1.00 (0.86–1.15)	-	Past surgical history
Yi KW [29]	2020	1,938,424	44,428	1,893,996	1.55 (1.52–1.58)	1.33 (1.30–1.35)	ICD-10 code
Porpora MG [30]	2020	425	145	280	1.20 (0.79–1.84)	-	Clinical or surgical diagnosis
Miura M [31]	2019	2769	80	2689	1.49 (0.96–2.33)	-	Clinical or surgical diagnosis
Uccella S [32]	2019	1808	118	1690	2.22 (1.52–3.26)	-	Past surgical history
Shmueli A [33]	2019	61,535	135	61,400	7.95 (5.58–11.31)	5.01 (3.34–7.52)	ICD-10 code
Nirgianakis K [34]	2018	248	62	186	1.79 (1.00–3.21)	-	Past surgical history
Chen I [35]	2018	52,202	469	51,733	1.39 (1.15–1.68)	1.08 (0.97–1.20)	ICD-10 code
Tzur T [36]	2018	502	35	467	50.83 (15.20–170.03)	38.08 (11.04–131.38)	Past surgical history
Li H [37]	2017	375	75	300	1.49 (0.89–2.50)	1.53 (0.83–2.84)	Past surgical history
Berlac JF [38]	2017	1,091,251	19,331	1,071,920	3.38 (3.27–3.50)	-	ICD-10 code
		1,075,846	3926 *	1,071,920	5.14 (4.81–5.50)	-	Past surgical history
Saraswat L [39]	2017	10,939	4232	6707	1.88 (1.72–2.05)	1.40 (1.26–1.55)	Past surgical history
Mannini L [40]	2017	786	262	524	2.32 (1.71–3.14)	-	Past surgical history
Glavind MT [42]	2017	82,793	1719	81,074	1.93 (1.72–2.16)	1.83 (1.60–2.09)	ICD-10 code
Harada T [41]	2016	9186	330	8856	1.61 (1.25–2.07)	-	Clinical or surgical diagnosis
Benaglia L [43]	2016	478	239	239	1.03 (0.72–1.48)	-	Clinical or surgical diagnosis
Exacoustos C [44]	2016	341	41	300	2.82 (1.40–5.65)	-	Past surgical history
Jacques M [45]	2016	223	110	113	2.64 (1.37–5.07)	-	Clinical or surgical diagnosis
Baggio S [46]	2015	123	30	93	3.67 (1.56–8.64)	-	Past surgical history
Lin H [47]	2015	498	249	249	2.45 (1.70–3.54)	1.93 (1.31–2.84)	Past surgical history
Conti N [48]	2015	2239	316	1923	1.17 (0.89–1.54)	-	Past surgical history
Mekaru K [49]	2014	88	40	48	1.62 (0.63–4.16)	-	Past surgical history
Benaglia L [50]	2012	191	61	130	1.18 (0.63–2.23)	1.25 (0.63–2.50)	Clinical diagnosis
Stephansson O [52]	2009	1,442,675	13,090	1,429,585	1.75 (1.68–1.83)	1.47 (1.40–1.54)	ICD-10 code
Kortelahti M [53]	2003	274	137	137	1.03 (0.62–1.73)	-	Past surgical history

* Restricted to women with histologically diagnosed endometriosis. Some values listed might differ slightly from the original values because the calculation was performed using Revman v5.4.1 or they were estimated by the authors. Abbreviations: CD, cesarean delivery; CI, confidence interval; Endo, endometriosis; ICD-10, International Classification of Diseases 10th Revision; No., number of cases; OR, odds ratio; -, not applicable.

**Table 3 biomedicines-10-00478-t003:** Comparator analysis of the CD rates between women with DE and non-DE.

Author	Year	Total No.	Endo No.	DE No.	Non-DE No.	Control No.	CD No. (%)		
							DE	Non-DE	Control
Uccella S [32]	2019	1808	118	34	84	1690	15 (44.1)	34 (40.5)	409 (24.2)
Nirgianakis K [34]	2018	248	62	62	-	186	36 (58.1)	-	81 (43.5)
Mannini L [40]	2017	120	40	40	-	80	18 (45.0)	-	23 (28.8)
Exacoustos C [44]	2016	341	41	41	-	300	28 (68.3)	-	130 (43.3)
Baggio S [46]	2015	123	30	30	-	93	18 (60.0)	-	27 (29.0)
Benaglia L [50]	2012	191	61	-	61 ^#^	130	-	23 (37.7)	44 (33.8)

^#^ In this study, all patients in the endometriosis group were patients with ovarian endometrioma. Some values listed might differ slightly from the original values because the calculation was performed using Revman v5.4.1 or they were estimated by the authors. Abbreviations: CD, cesarean delivery; DE, deep endometriosis; Endo, endometriosis; No., number of cases; -, not applicable.

**Table 4 biomedicines-10-00478-t004:** Comparator analysis of the rates of PPH between women with and without endometriosis.

Author	Year	No.	Endo No.	Control No.	PPH		Definition of Endo	Definition of PPH
					OR (95%CI)	Adjusted OR (95%CI)		
Lin S [27]	2020	246	82	164	1.18 (0.45–3.13)	-	Hist	-
Yi KW [29]	2020	1,938,424	44,428	1,893,996	1.12 (1.09–1.16)	1.10 (1.07–1.14)	ICD-10 code	ICD-10 code
Porpora MG [30]	2020	425	145	280	2.62 (0.58–11.86)	-	Clinical, hist	500 mL VD 1000 mL CD
Miura M [31]	2019	2769	80	2689	1.70 (1.03–2.81)	-	Clinical, hist	800 mL VD 1500 mL CD
Uccella S [32]	2019	1808	118	1690	0.67 (0.41–1.09)	-	Hist	-
Shmueli A [33]	2019	61,535	135	61,400	2.40 (1.06–5.46)	3.69 (1.60–8.53)	ICD-10 code	500 mL VD 1000 mL CD
Nirgianakis K [34]	2018	248	62	186	2.04 (0.87–4.78)	-	Hist	500 mL VD 1000 mL CD
Chen I [35]	2018	52,202	469	51,733	1.10 (0.77–1.55)	1.02 (0.74–1.41)	ICD-10 code	ICD-10 code
Tzur T [36]	2018	502	35	467	1.87 (0.09–36.90)	-	Hist	-
Li H [37]	2017	375	75	300	2.18 (1.16–4.09)	2.27 (1.06–4.87)	Hist	-
Berlac JF [38]	2017	1,091,251	19,331	1,071,920	0.86 (0.82–0.90)	0.95 (0.90–1.00)	ICD-10 code	-
		1,075,846	3926 *	1,071,920	1.45 (1.33–1.59)	0.87 (0.80–0.95)	Hist	-
Glavind MT [42]	2017	82,793	1719	81,074	0.98 (0.83–1.16)	0.95 (0.80–1.14)	ICD-10 code	500 mL
Saraswat L [39]	2017	10,939	4232	6707	1.88 (1.69-2.09)	1.37 (1.16–1.61)	Hist	500 mL VD 1000 mL CD
Mannini L [40]	2017	786	262	524	0.89 (0.52–1.50)	-	Hist	500 mL
Jacques M [45]	2016	223	113	113	1.00 (0.40–2.50)	-	Clinical, hist	-
Conti N [48]	2015	2239	316	1923	1.13 (0.75–1.69)	-	Hist	500 mL VD 750 mL CD
Healy DL [51]	2010	6730	1265	5465	1.26 (1.05–1.52)	1.29 (1.06–1.56)	-	500 mL VD 750 mL CD

* Restricted to women with histologically diagnosed endometriosis. Some values listed might differ slightly from the original values because the calculation was performed using Revman v5.4.1 or they were estimated by the authors. Abbreviations: CD, cesarean delivery; CI, confidence interval; Endo, endometriosis; Hist, histologically confirmed endometriosis; ICD-10, International Classification of Diseases 10th Revision; OR, odds ratio; PPH, postpartum hemorrhage; No., number of cases; VD, vaginal delivery; -, not applicable.

## Data Availability

All the studies included in this study are published in the literature.

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
