# Peer review of "Association between Endometriosis and Delivery Outcomes: A Systematic Review and Meta-Analysis"

_biomedicines, 2022, doi:10.3390/biomedicines10020478_

Round 1

Reviewer 1 Report

I read with interest a paper on important issues in reproductive medicine. Unfortunately, the authors did not avoid a number of shortcomings that need to be corrected. The work seems to be valuable and the obtained results are interesting, while some conclusions seem to be unauthorized (do not result from the presented data) or even led astray. I will present the most important remarks in the points below:

1.

The summary begins by stating: "While the detection rate of endometriosis in pregnant women is increasing due to advanced assisted reproductive technologies, their delivery outcomes are understudied."
and in the first paragraph we find the sentence:
"Nevertheless, recent advances in assisted reproductive technology have led to improved pregnancy rates in women with endometriosis"

The first thesis is illogical because assisted reproductive methods are not used to „detect” endometriosis, nor is endometriosis „detected”at birth. Additionally, not all cases of endometriosis are associated with any fertility impairment (majority are fertile [4]), although statistically there is some accumulation of endometriosis cases among infertile patients. According to the WES, ESHRE and ASRM guidelines, patients with endometriosis can be treated surgically and IVF treatment is only one of the options. Additionally, in recent years there have been many studies showing that the results of surgical treatment or alternative approaches can be quite effective, even when compared to assisted reproductive methods. Already 10 years ago, Adamson et al, when publishing the Endometriosis Fertility Index (EFI), indicated that up to 60-80% of patients may become pregnant spontaneously within 3 years after endometriosis surgery. The lower efficiency resulted mainly from other co-related problems than endometriosis. Additionally, Johnson by publishing the results of FLUSH study PMID: 17890725 showed that as a result of a simple intervention such as HSG, up to 60% of patients with endometriosis can become pregnant within 2 years.

In justifying thesis 2, the authors refer to the work of Senapati 2016. The work refers to historical data from over 10 years ago and, unfortunately, does not say anything about the fact that IVF improvements changed the effectiveness of infertility treatment in endometriosis. Was IVF less effective in the past? Do we have any evidence of this? Do we know what percentage of delivering women with endometriosis were treated with 1)IVF, 2)surgery 3)conceived spontaneously without treatment? In order to formulate such theses as above, it would be necessary to answer these important questions. If we don't have an answer, it's better not to make any speculative theses.

2.

There is a controversial sentence in the first paragraph:
"The number of patients with endometriosis has increased in recent years, accounting for about 5% −15% of reproductive-aged women and approximately 25% −50% of infertile women", which is justified by the Eisenberg 2018 publication. Interestingly, this publication does not say anything about the change in the frequency of endometriosis in recent times, and even indicates that the incidence of endometriosis is lower than in previously published reports (sic!)! Eisenberg indicates an incidence ranging from 1 to 1.8%, so why is his work cited when you indicate an incidence of 5-15% ???

3.

The reviewer strongly suggests using the term Deep endometriosis (DE) instead of DIE. In the latest publications (from 2020) this term is preferred, e.g. PMID: 32064361. The term DE is more correct as the term DIE incorrectly assumes that endometriosis nodules "infiltrate", which even if it happens at all, certainly does not happen in all cases DE.

4.

In paragraph 2.2, there is the following sentence:
"In this study, patients with severe endometriosis were those with deep invasive endometriosis (DIE) or revised American Society of Reproductive Medicine (rASRM) stage III − IV endometriosis, whereas patients with non-severe endometriosis were those with non-DIE or rASRM stage I − II endometriosis. "
Classifications of endometriosis are the subject of much controversy, but it should be noted that the role of Deep Endometriosis is ambiguous in terms of fertility and reproduction. The rASRM and EFI index do not take into account the presence of DE in its staging and it is not fully known whether it has an impact on fertility. It is the authors' right to designate "severe endometriosis" as they see fit, but the limitations and controversies associated with such selection should be noted.

5.

I find the part regarding midline incision and uterine exteriorization unnecessary. As the authors rightly point out in 4.3.1, this is based on speculation and there is no evidence of drawing conclusions for practical purposes. The authors themselves cite the work of Maaløe et al 2014, where it was found that a midline incision is a typical technique for low-income countries, in highly developed countries almost exclusively the transverse technique is used. Maaloe et al advocate a shift from the midline to the transverse to avoid unnecessary complications also in low-income countries. The midline technique is commonly treated as part of the history of medicine.
Furthermore, I do not know of any evidence that uterine exteriorization is used as a routine technique in cesarean section. I run the Ob & G department in a large European city where we have a large endometriosis center and nearly 7,000 births a year. The only indication for a midline incision I allow is an old midline scar from previous surgery. If in exceptional cases, there is a need to significantly expand the operating field, I make an additional midline incision. It happens extremely, rarely, and I believe that it is better to make an additional incision in 1 patient than to expose 100 patients to the planned complications of midline incision performed unnecessarily. I also have to say that the performance of a planned midline incision would be considered malpractice by most department heads I know, and patients would bring legal claims for damages.

Author Response

Thank you for reviewing our paper and for considering it for publication in Biomedicines. We have reviewed our manuscript carefully and have made extensive revisions to meet the reviewers’ suggestions and requirements.

Reviewer #1 comments:

I read with interest a paper on important issues in reproductive medicine. Unfortunately, the authors did not avoid a number of shortcomings that need to be corrected. The work seems to be valuable and the obtained results are interesting, while some conclusions seem to be unauthorized (do not result from the presented data) or even led astray. I will present the most important remarks in the points below:

Thank you for your helpful comments. We have provided point-by-point responses to all your comments and have also provided explanations for revisions made to the manuscript.

Reviewer #1, Comment 1:

The summary begins by stating: "While the detection rate of endometriosis in pregnant women is increasing due to advanced assisted reproductive technologies, their delivery outcomes are understudied."

And in the first paragraph we find the sentence: "Nevertheless, recent advances in assisted reproductive technology have led to improved pregnancy rates in women with endometriosis"

The first thesis is illogical because assisted reproductive methods are not used to “detect” endometriosis, nor is endometriosis “detected” at birth. Additionally, not all cases of endometriosis are associated with any fertility impairment (majority are fertile [4]), although statistically there is some accumulation of endometriosis cases among infertile patients. According to the WES, ESHRE and ASRM guidelines, patients with endometriosis can be treated surgically and IVF treatment is only one of the options. Additionally, in recent years there have been many studies showing that the results of surgical treatment or alternative approaches can be quite effective, even when compared to assisted reproductive methods. Already 10 years ago, Adamson et al, when publishing the Endometriosis Fertility Index (EFI), indicated that up to 60-80% of patients may become pregnant spontaneously within 3 years after endometriosis surgery. The lower efficiency resulted mainly from other co-related problems than endometriosis. Additionally, Johnson by publishing the results of FLUSH study PMID: 17890725 showed that as a result of a simple intervention such as HSG, up to 60% of patients with endometriosis can become pregnant within 2 years.

In justifying thesis 2, the authors refer to the work of Senapati 2016. The work refers to historical data from over 10 years ago and, unfortunately, does not say anything about the fact that IVF improvements changed the effectiveness of infertility treatment in endometriosis. Was IVF less effective in the past? Do we have any evidence of this? Do we know what percentage of delivering women with endometriosis were treated with 1)IVF, 2)surgery 3)conceived spontaneously without treatment? In order to formulate such theses as above, it would be necessary to answer these important questions. If we don't have an answer, it's better not to make any speculative theses.

Reply: Abstract; and Page 1, paragraph 1

We sincerely appreciate these valuable comments. We completely agree with the reviewer’s comments and noticed that we misunderstood the relationship between endometriosis and assisted reproductive technology (ART). Moreover, the suggested studies were very valuable to us. As the reviewer has pointed out, the description regarding the relationship between endometriosis and ART is inappropriate; thus, we have deleted the descriptions accordingly.

Reviewer #1, Comment 2:

There is a controversial sentence in the first paragraph:

"The number of patients with endometriosis has increased in recent years, accounting for about 5% −15% of reproductive-aged women and approximately 25% −50% of infertile women", which is justified by the Eisenberg 2018 publication. Interestingly, this publication does not say anything about the change in the frequency of endometriosis in recent times, and even indicates that the incidence of endometriosis is lower than in previously published reports (sic!)! Eisenberg indicates an incidence ranging from 1 to 1.8%, so why is his work cited when you indicate an incidence of 5-15% ???

Reply: Page 1, paragraph 1

Thank you for your helpful comments. As the reviewer has pointed out, the cited studies are not appropriate; therefore, we have added and revised the references regarding the incidence of endometriosis [1-4].

Reviewer #1, Comment 3:

The reviewer strongly suggests using the term Deep endometriosis (DE) instead of DIE. In the latest publications (from 2020) this term is preferred, e.g. PMID: 32064361. The term DE is more correct as the term DIE incorrectly assumes that endometriosis nodules "infiltrate", which even if it happens at all, certainly does not happen in all cases DE.

Reply: Page 3, paragraph 3; (and throughout the manuscript thereafter)

Thank you for your helpful insight. We have changed the term DIE to DE throughout the manuscript.

Reviewer #1, Comment 4:

In paragraph 2.2, there is the following sentence:

"In this study, patients with severe endometriosis were those with deep invasive endometriosis (DIE) or revised American Society of Reproductive Medicine (rASRM) stage III − IV endometriosis, whereas patients with non-severe endometriosis were those with non-DIE or rASRM stage I − II endometriosis. "

Classifications of endometriosis are the subject of much controversy, but it should be noted that the role of Deep Endometriosis is ambiguous in terms of fertility and reproduction. The rASRM and EFI index do not take into account the presence of DE in its staging and it is not fully known whether it has an impact on fertility. It is the authors' right to designate "severe endometriosis" as they see fit, but the limitations and controversies associated with such selection should be noted.

Reply: Table 3; Figure 5; and Page 11, paragraph 1

We appreciate these valuable comments, which we completely agree with. We have deleted the classification concerning the severity of endometriosis according to the reviewer’s suggestion. To report findings on delivery outcomes for patients with DE, we have added the results of a comparative analysis of cesarean delivery rates between women with DE versus women without DE.

Reviewer #1, Comment 5:

I find the part regarding midline incision and uterine exteriorization unnecessary. As the authors rightly point out in 4.3.1, this is based on speculation and there is no evidence of drawing conclusions for practical purposes. The authors themselves cite the work of Maaløe et al 2014, where it was found that a midline incision is a typical technique for low-income countries, in highly developed countries almost exclusively the transverse technique is used. Maaloe et al advocate a shift from the midline to the transverse to avoid unnecessary complications also in low-income countries. The midline technique is commonly treated as part of the history of medicine.

Furthermore, I do not know of any evidence that uterine exteriorization is used as a routine technique in cesarean section. I run the Ob & G department in a large European city where we have a large endometriosis center and nearly 7,000 births a year. The only indication for a midline incision I allow is an old midline scar from previous surgery. If in exceptional cases, there is a need to significantly expand the operating field, I make an additional midline incision. It happens extremely, rarely, and I believe that it is better to make an additional incision in 1 patient than to expose 100 patients to the planned complications of midline incision performed unnecessarily. I also have to say that the performance of a planned midline incision would be considered malpractice by most department heads I know, and patients would bring legal claims for damages.

Reply: This section has now been deleted from the manuscript

Thank you very much for your helpful comments. We sincerely agree with your opinion and have removed that consideration from the manuscript.

Reviewer 2 Report

Thank you very much for allowing me to review your paper. I was very interested in your methodology and systematic survey of papers on endometriosis and pregnancy outcomes. However, before accepting the paper, I would like to raise a few concerns.

1. First, in a systematic review, it would be better to adjust for age as a bias in each study. Each of the research papers discussed in this review has a different age distribution. I think it is natural that there are differences in patient backgrounds among the papers, but is it not necessary to adjust for baseline? In particular, does age and the presence of endometriosis have any effect on PPH rates? I also thought that adenomyosis and ovarian endometriotic cysts seem to affect the PPH rate differently, do we need to explain that?

2. Such a meta-analysis requires registration in PROSPERO. Have the authors registered?

3. The authors state that a midline incision is preferable for patients with endometriosis, but is it essential to elevate the uterus so far ventrally? Is it essential to elevate the uterus so far ventrally? In the case of flaccid bleeding, it is certainly necessary to grasp the uterus firmly, so a midline incision is better because it allows the operative field to expand, but is it difficult to cut the fascia vertically only in that case? Also, I heard that endometriosis sometimes improves during pregnancy. How do you evaluate that? The Matsubara-Neraton method seems to be the way to go, but I don't think a midline vertical incision is necessary in all cesarean cases.

Author Response

Thank you for reviewing our paper and for considering it for publication in Biomedicines. We have reviewed our manuscript carefully and have made extensive revisions to meet the reviewers’ suggestions and requirements.

Reviewer #2 comments:

Thank you very much for allowing me to review your paper. I was very interested in your methodology and systematic survey of papers on endometriosis and pregnancy outcomes. However, before accepting the paper, I would like to raise a few concerns.

Thank you for your helpful comments. We have provided point-by-point responses to all your comments and have also provided explanations for the revisions made in the manuscript.

Reviewer #2, Comment 1:

First, in a systematic review, it would be better to adjust for age as a bias in each study. Each of the research papers discussed in this review has a different age distribution. I think it is natural that there are differences in patient backgrounds among the papers, but is it not necessary to adjust for baseline? In particular, does age and the presence of endometriosis have any effect on PPH rates? I also thought that adenomyosis and ovarian endometriotic cysts seem to affect the PPH rate differently, do we need to explain that?

Reply: Page 17, paragraph 3; and Page 18, paragraph 7

Thank you for your helpful comments. Several studies that examine the effect of endometriosis on the rate of cesarean delivery have used multivariate analysis to adjust for confounding factors such as age, parity, and BMI. For example, adjusted OR was used to calculate adjusting for age in Berlac et al.’s study [5]; and for age, parity, socio-economic status, and year of pregnancy in a study by Saraswat L, et al. [6] using multivariate analysis. In the adjusted analysis of meta-analysis, the adjusted odds ratio from the results of multivariate analysis were combined. Therefore, some confounding factors concerning cesarean delivery would be excluded in the adjusted analysis. However, we are unable to exclude all of confounding factors; thus, we emphasize this point as a notable limitation of this study.

As you have pointed out, a recent comprehensive review reported that advanced maternal age was a risk factor for PPH [7]. Other risk factors for PPH include obstetric complications such as placenta previa and maternal complications such as uterine myoma. Many of the studies described adjusted OR in this systematic review were adjusted for maternal age, but not for various other confounding factors. Therefore, this study cannot clearly conclude that endometriosis is a risk factor for PPH as other potential confounding factors had been excluded. These points have been added to the limitations section.

Only one of the 28 included studies had evaluated the risk of PPH in patients with ovarian endometrioma whose endometriotic lesions were confined to the ovaries [8]. That study reported that the risk of PPH was similar between patients with and without ovarian endometrioma (15.6% [10/64] vs. 24.4% [413/1,690], p = 0.13) [8]. These findings from only one report need to be supplemented through further research to fully determine the association between ovarian endometrioma and PPH. We have noted this concern in Discussion section of the manuscript.

Reviewer #2, Comment 2:

Such a meta-analysis requires registration in PROSPERO. Have the authors registered?

Reply: Page 18, paragraph 6

Thank you for your helpful suggestions. Please also refer to Editor’s Comment.

PROSPERO registration is required upon completion of data extraction (Guidance notes for registering a systematic review protocol with PROSPERO: https://www.crd.york.ac.uk/prospero/documents/Registering%20a%20review%20on%20PROSPERO.pdf; page 3, last paragraph). Therefore, we cannot register this systematic review at this phase. We completely agree with the reviewer’s suggestion and will register future studies in PROSPERO, but please understand that the current study cannot be registered in PROSPERO. Without preregistration, it is not known whether the main outcomes, such as the rate of cesarean delivery, were predefined as primary outcomes. Therefore, this may have biased the findings of our systematic review and we have noted this point as a limitation of this study.

Reviewer #2, Comment 3:

The authors state that a midline incision is preferable for patients with endometriosis, but is it essential to elevate the uterus so far ventrally? Is it essential to elevate the uterus so far ventrally? In the case of flaccid bleeding, it is certainly necessary to grasp the uterus firmly, so a midline incision is better because it allows the operative field to expand, but is it difficult to cut the fascia vertically only in that case? Also, I heard that endometriosis sometimes improves during pregnancy. How do you evaluate that? The Matsubara-Neraton method seems to be the way to go, but I don't think a midline vertical incision is necessary in all cesarean cases.

Reply: Deleted from the text

Thank you very much for your helpful comments. Please also refer to Reviewer #1, Comment #5.

Our considerations and suggestions were based on our experience in high-risk cases for postpartum hemorrhage such as placenta previa and, as you have pointed out, were inappropriate for consideration in all of the pregnant women with endometriosis. We agree with your opinion and we have removed that consideration from the manuscript.

Round 2

Reviewer 1 Report

The work may be published in its current form. Thank you for taking into account the amendments.

Reviewer 2 Report

Dear Authors,

Thank you very much for your reply. The authors responded to all my concerns. 

Sincerely,